

# Conjectures about the structure of strong- and weak-coupling expansions of a few ground-state observables in the Lieb-Liniger and Yang-Gaudin models

**Guillaume Lang**

CNRS, LPMMC, F-38000 Grenoble, France

## Abstract

In this paper, we apply experimental number theory to two integrable quantum models in one dimension, the Lieb-Liniger Bose gas and the Yang-Gaudin Fermi gas with contact interactions. We identify patterns in weak- and strong-coupling series expansions of the ground-state energy, local correlation functions and pressure. Based on the most accurate data available in the literature, we make a few conjectures about their mathematical structure and extrapolate to higher orders.



# 1  Introduction

The Lieb-Liniger model describes spinless bosons with contact interactions, whose motion is confined to one dimension [1]. It is arguably the most conceptually simple, integrable model in the continuum. Each of its observables is amenable to exact calculations using Bethe Ansatz techniques [1, 2], providing valuable insights in the properties of strongly-correlated systems in one dimension [3]. The accuracy of the ground-state solution can be used to benchmark effective theories, such as the Luttinger liquid approximation [4–6] and its non-linear generalization [7, 8], as well as numerical techniques [9–12]. Last, but not least, this model can be experimentally realized in ultracold gases experiments, both in the repulsive [13–18] and attractive regime [19].

Its fermionic, spin-1/2 counterpart, known as the Yang-Gaudin model [20, 21], and its generalization to arbitrary spin [22], has attracted increasing attention [23–26] since it has been experimentally realized up to $\kappa = 6$ spin components [27] in an harmonic trap. Interestingly, the spin-1/2 model with repulsive interactions is related to the Lieb-Liniger model with attractive interactions [28], known as the super-Tonks-Girardeau gas [29, 30], and the large-spin limit coincides with the Lieb-Liniger model with repulsive interactions [31]. A vast web of mappings, onto the Kardar-Parisi-Zhang (KPZ) model [32] or directed polymers [33] for instance, and the universality of Lieb-Liniger-like models as non-relativistic limits in one dimension [34, 35], make the open problem of their full explicit resolution even more compelling.

A key observable is the ground-state energy, which is related to many other physical quantities through theorems and thermodynamic relations. For instance, in the Lieb-Liniger model, the local two-body correlation function can be extracted from the latter [36], as well as the Luttinger coefficients [5] that govern the long-range behaviour of non-local correlation functions, and Tan's contact, that yields the high-momentum tail of the momentum distribution [37–39]. Most remarkably, even the excitation spectrum can be obtained [40], by calculating the effective mass [41] and higher-order coefficients of its low-momentum expansion [42]. This spectrum, in turn, is related to the dynamical structure factor [43].

Although the exact ground-state energy can be obtained, in principle, from the exact Bethe Ansatz equations, only weak- and strong-coupling expansions are accessible to date [1, 40, 44–51]. As far as the Lieb-Liniger model is concerned, recently-developed algorithms allow to obtain these expansions to any order [40, 49, 50]. An important step forward would be to understand them well enough to predict their coefficients at arbitrary order without evaluating them algorithmically.

This paper is organized as follows. In section 2, we briefly describe the Lieb-Liniger and Yang-Gaudin models, introduce a few notations and the Bethe Ansatz equations that give access to the ground-state energy. In section 3, we sum up the main known results about their strong- and weak-coupling expansions. We then propose new conjectures about the ground-state energy of the Yang-Gaudin model, and discuss the radius of convergence of the partial resummations we perform. In section 4, we sum up the main analytical developments related to the local correlation functions of the Lieb-Liniger model and propose a conjecture about their structure. Then, in section 5, we link together two independent descriptions of the local two-body correlation function to evaluate an observable that we identify as the pressure of the Bose gas. We provide conjectures about its strong-coupling expansion. In section 6, we conclude and give a few outlooks.

## 2 Models and equations

The one-dimensional quantum gas composed of $N$ identical spinless point-like bosons of mass $m$ with contact interactions, is described by the Lieb-Liniger Hamiltonian $H$ which reads [1]

$$H = \sum_{i=1}^{N} \left[ -\frac{\hbar^2}{2m} \frac{\partial^2}{\partial x_i^2} + \frac{g_{1D}}{2} \sum_{j \neq i} \delta(x_i - x_j) \right], \tag{1}$$

where $\{x_i\}_{i \in \{1,\dots,N\}}$ label the positions of the bosons, $\hbar$ is the Planck constant divided by $2\pi$, $g_{1D}$ is an effective one-dimensional coupling constant and $\delta$ is the Dirac function.

It is customary to introduce a dimensionless coupling constant, known as Lieb's parameter, that reads

$$\gamma = \frac{m g_{1D}}{n_0 \hbar^2}, \tag{2}$$

where $n_0$ is the average linear density.

The coordinate Bethe Ansatz procedure allowing to solve this model is well-known, see e.g. [52] for details. It yields a set of three equations, namely

$$g(z; \alpha) - \frac{1}{2\pi} \int_{-1}^{1} dy \frac{2\alpha g(y; \alpha)}{\alpha^2 + (y - z)^2} = \frac{1}{2\pi}, \tag{3}$$

where $g(z; \alpha)$ denotes the distribution of quasi-momenta in reduced units, $z$ is the pseudo-momentum in reduced units such that its maximal value is 1 and $\alpha$ is a real number in one-to-one correspondence with the Lieb parameter $\gamma$ introduced above through a second equation,

$$\gamma \int_{-1}^{1} dy\, g(y; \alpha) = \alpha. \tag{4}$$

A third equation yields the dimensionless average ground-state energy per particle, $e_B(\gamma)$, linked to the total energy $E_0$ by

$$e_B(\gamma) = \frac{2m}{\hbar^2} \frac{E_0(\gamma)}{N n_0^2}, \tag{5}$$

according to

$$e_B(\gamma) = \frac{\int_{-1}^{1} dy\, g(y; \alpha(\gamma)) y^2}{[\int_{-1}^{1} dy\, g(y; \alpha(\gamma))]^3}. \tag{6}$$

A similar procedure allows to obtain the ground-state energy of the Yang-Gaudin model of spin-1/2 fermions with contact interactions, and even the $\kappa$-component model with an arbitrary number of spin components, see e.g. [53, 54] for details. The Hamiltonian stays the same as in Eq. (1), but the statistics is modified to take into account the fermionic nature of the particles. In the case of a $\kappa$-component Fermi gas divided into $[m_1, \dots, m_\kappa]$ particles per species, a set of $\kappa$ rapidities, distributions and integrations bounds denoted respectively $k_i$, $\rho_i(k_i)$ and $B_i$ can be defined. Introducing the notation $M_i = \sum_{j=i}^{\kappa} m_j$ for all $i \in \{1, \dots, \kappa\}$, the $\kappa$ coupled integral Bethe Ansatz equations for the ground state in the thermodynamic limit can be written as:

$$2\pi \rho_1 = 1 + 4g_{1D} \int_{-B_2}^{B_2} \frac{\rho_2(k_2) dk_2}{g_{1D}^2 + 4(k_1 - k_2)^2}, \tag{7}$$

$$4g_{1D}\int_{-B_{i+1}}^{B_{i+1}}\frac{\rho_{i+1}(k_{i+1})dk_{i+1}}{g_{1D}^2+4(k_i-k_{i+1})^2}+4g_{1D}\int_{-B_{i-1}}^{B_{i-1}}\frac{\rho_{i-1}(k_{i-1})dk_{i-1}}{g_{1D}^2+4(k_i-k_{i-1})^2}$$
$$=2\pi\rho_i+2g_{1D}\int_{-B_i}^{B_i}\frac{\rho_i(k_i')dk_i'}{g_{1D}^2+4(k_i-k_{i'})^2} \tag{8}$$

and

$$4g_{1D}\int_{-B_{\kappa-1}}^{B_{\kappa-1}}\frac{\rho_{\kappa-1}(k_{\kappa-1})dk_{\kappa-1}}{g_{1D}^2+4(k_\kappa-k_{\kappa-1})^2}=2\pi\rho_\kappa+2g_{1D}\int_{-B_\kappa}^{B_\kappa}\frac{\rho_\kappa(k_\kappa')dk_\kappa'}{g_{1D}^2+4(k_\kappa-k_{\kappa'})^2}, \tag{9}$$

together with the following normalization condition:

$$\frac{M_i}{L}=\int_{-B_i}^{B_i}\rho_i dk_i. \tag{10}$$

The dimensionless average ground-state energy per particle reads:

$$e_F(\gamma;\kappa)=\int_{-B_1}^{B_1}\rho_1 k_1^2 dk_1. \tag{11}$$

## 3 Ground-state energy

Although solving the Lieb equations yields, in principle, the exact ground-state energy, only weak and strong-coupling expansions are known to date. In this section, we recap the most accurate results available and take a close look at their patterns to try and identify their generating series.

### 3.1 Strong-coupling expansion

Neumann series expansion of the kernel is the standard mathematical tool to solve integral equations like Eq. (3), see e.g. [46]. However, only the two first orders of this series expansion have been derived analytically [40], due to the increasing complexity of higher-order terms.

An algorithm that yields systematic series expansions at large Lieb parameter $\gamma$ has been developed in [40]. It allows to evaluate analytically the ground-state energy of the Lieb-Liniger model in the vicinity of the Tonks-Girardeau regime $\gamma\to+\infty$ as a truncated series in the inverse Lieb parameter,

$$e_B(\gamma)=\sum_{k=0}^{n}\frac{a_k}{\gamma^k}+O\left(\frac{1}{\gamma^{n+1}}\right). \tag{12}$$

We have pushed this procedure up to order $n=20$ in [55], the explicit result is given here in appendix A. More generally, at order $n$ the expression contains $1+\lfloor(n+1)/2\rfloor\lfloor 1+n/2\rfloor$ different terms, where $\lfloor\dots\rfloor$ is the floor function, such that $\lfloor x\rfloor$ is the nearest lower integer to $x$, or $x$ if the latter is integer. For this reason, explicit expressions of $a_k$ are rather uncanny for large $n$, all the more so as the coefficients become more and more complex. Moreover, the convergence of the series to the exact value at given $\gamma$ is quite slow at intermediate coupling, and the procedure only applies when $\alpha>2$ in Eq. (3), corresponding to $\gamma>4.527$.

The will to bypass these shortcomings led us to look at the problem from the perspective of experimental number theory, trying to identify patterns in the expansion to express it with more compact notations. This heuristic point of view turned out to be rather fruitful.

We guessed a structure, seemingly valid at all orders, whose partial resummation led to the following pattern:

$$e_B(\gamma) = \sum_{n=0}^{+\infty} e_{B,n}(\gamma), \tag{13}$$

where

$$e_{B,0}(\gamma) = \frac{\pi^2}{3} \frac{\gamma^2}{(2+\gamma)^2} \tag{14}$$

and for $n \geq 1$,

$$e_{B,n}(\gamma) = \frac{\gamma^2}{(2+\gamma)^{3n+2}} \pi^{2n+2} \mathcal{E}_n(\gamma), \tag{15}$$

where $\mathcal{E}_n(\gamma) = \sum_{k=0}^{n-1} a_{k;n} \gamma^k$ is a polynomial of degree $n-1$ with rational coefficients. We also guessed a few properties of the latter. In particular, we identified the coefficient of the the highest-degree monomial of $\mathcal{E}_n$ [55]. This conjecture generalizes a result of [56] to arbitrary order, and is in agreement with the exact series expansion to order 20. The role of the factor $1 + \frac{2}{\gamma}$ that appears at the denominator when factorizing by $\gamma$ is discussed in an appendix of [57]. Other lines could be followed to seek structures, see Appendix B for an example.

The validity of these assumptions has been checked at higher orders by independent numerical calculations [58]. Identifying the structure of coefficients in the polynomials $\mathcal{E}_n$ allows to make more accurate calculations [55], compared to the mere series expansion in $1/\gamma$. Indeed, a major advantage of this partially resummed expansion is that its radius of convergence is infinite. A closed-form formula for the coefficients that would lead to a full resummation would solve the ground-state energy problem explicitly.

In comparison, relatively few results have been obtained concerning the Yang-Gaudin and general $\kappa$-component model, due to the much more complex structure of the equations involved. In the balanced gas where all spin sectors contain the same number of fermions, the most accurate strong-coupling expansion of the average ground-state energy per particle available to date reads [60]

$$\frac{e_F(\gamma; \kappa)}{\pi^2/3} = 1 - \frac{4Z_1(\kappa)}{\gamma} + \frac{12Z_1(\kappa)^2}{\gamma^2} - \frac{32}{\gamma^3}\left(Z_1(\kappa)^3 - \frac{Z_3(\kappa)\pi^2}{15}\right) + O\left(\frac{1}{\gamma^4}\right), \tag{16}$$

where $Z_1(\kappa) = -\frac{1}{\kappa}\left(\psi\left(\frac{1}{\kappa}\right) + C\right)$ and $Z_3(\kappa) = \frac{1}{\kappa^3}\left(\zeta\left(3, \frac{1}{\kappa}\right) - \zeta(3)\right)$, with $\psi$ the Euler psi function, $C$ the Euler constant and $\zeta$ the Riemann zeta function.

This result generalizes the one obtained previously for the Yang-Gaudin model in [61], since $Z_1(\kappa = 2) = \ln(2)$ and $Z_3(\kappa = 2) = \frac{3}{4}\zeta(3)$. It also agrees with its well-kwown Lieb-Liniger counterpart through Yang and You's generalized Lieb-Mattis theorem for $\kappa \to \infty$ [31] that states the equivalence between spinless bosons and balanced infinite-spin fermions in the thermodynamic limit, as $Z_1(\kappa) \to_{\kappa \to \infty} 1$ and $Z_3(\kappa) \to_{\kappa \to \infty} 1$.

Inspired by the results presented in this subsection, we state a conjecture about the structure of the series expansion of the $\kappa$-component fermi gas.

**Conjecture I:** The resummation above, Eq. (13), suggests a similar structure for $e_B(\gamma)$ and $e_F(\gamma; \kappa)$ through the Yang-You theorem as:

$$e_F(\gamma; \kappa) = \sum_{n=0}^{+\infty} e_{F,n}(\gamma; \kappa), \tag{17}$$

where

$$e_{F,0}(\gamma;\kappa) = \frac{\pi^2}{3} \frac{\gamma^2}{(2Z_1(\kappa)+\gamma)^2} \tag{18}$$

is inferred from the third-order strong-coupling expansion above.

It might be that $2Z_1(\kappa)+\gamma$ plays the role of $2+\gamma$ at the denominator to all orders, and that linear combinations of 1 and terms of the form $Z_{2j+1}(\kappa)$ appear as prefactors at the numerator, but too few orders are available to date to make reliable conjectures about the higher-order structures $e_{F,n}$ when $n \geq 1$.

## 3.2 Weak-coupling expansion

In the weakly-interacting regime, expansions of the ground-state energy of the Lieb-Liniger model have the following structure [46]:

$$e_B(\gamma) = \gamma\left(\sum_{k=0}^{n} c_k^B \gamma^{k/2} + O(\gamma^{(n+1)/2})\right). \tag{19}$$

The zeroth- and first-order coefficients have been obtained in [1]. The second-order term was inferred in [44] and checked wih more rigor in [45, 46]. Coefficients of order 3 to 5 were inferred from a very accurate numerical study [47, 48], and involve the zeta function evaluated at odd arguments, $\zeta(3)$ and $\zeta(5)$.

Very recently, further coefficients have been explicitly obtained using two independent methods. On the one hand, an efficient algorithmic expansion at low $\gamma$, combined to an integer coefficients algorithm to find the explicit form of the latter has been used in [49] and allowed to identify coefficients up to order 8. On the other hand, a systematic algorithmic procedure that yields directly the exact coefficients has been developed in [50], and has yielded their value up to order 34, though they have been published up to eighth order only due to their lengthy expressions. These studies revealed that from a certain order on, $c_k^B$ involves products of the zeta function evaluated at odd arguments. The asymptotic expression of $c_k^B$ for $k \gg 1$ has even been inferred from numerical data [50].

As far as the Yang-Gaudin model is concerned, the weak-coupling expansion of the ground-state energy reads

$$e_F(\gamma;\kappa=2) = \sum_{k=0}^{n} c_k^F(\kappa=2)\gamma^k + O(\gamma^{n+1}). \tag{20}$$

For a long time, only the first three coefficients were known [62]. A few more have been obtained numerically in [47, 48], and the fourth has been computed explicitly [51]. A systematic procedure that yields the exact coefficients has been developed [50, 51], and has given the explicit form of the coefficients up to order 34. They are given explicitly up to order 10 in [51].

We have carefully studied these new available data and found out structures that may be valid at arbitrary order. We thus make a few conjectures about subseries in the weak-coupling expansion of the Yang-Gaudin model.

**Conjecture II:** We guess that $\zeta(3)$ appears as a global factor in a peculiar subsequence of the weak-coupling expansion of the ground-state energy of the Yang-Gaudin model, as:

$$
\begin{aligned}
e_{F;\zeta(3)}(\gamma;\kappa=2) &= -\zeta(3)\sum_{n\geq 0}\frac{1}{2^{n+1}}(n+2)\binom{2n}{n}\frac{\gamma^{n+3}}{\pi^{2(n+2)}} \\
&= -\frac{\zeta(3)}{\pi^4}\gamma^3 - \frac{3\zeta(3)}{2\pi^6}\gamma^4 - \frac{3\zeta(3)}{\pi^8}\gamma^5 - \frac{25\zeta(3)}{4\pi^{10}}\gamma^6 - \frac{105\zeta(3)}{8\pi^{12}}\gamma^7 \dots \quad (21)
\end{aligned}
$$

This result is in full agreement with [51] to all published orders. It has been confirmed up to order 50 by the authors of the aforementioned article [59]. This subseries converges provided that $\gamma \leq \pi^2/2$, and sums up to

$$
e_{F;\zeta(3)}(\gamma;\kappa=2) = -\frac{\zeta(3)\gamma^3}{\pi^4}\frac{1-\frac{3}{2}\frac{\gamma}{\pi^2}}{\left[1-2\frac{\gamma}{\pi^2}\right]^{3/2}}. \quad (22)
$$

**Conjecture III:** We also point out that $\zeta(5)$ appears as a global factor in a peculiar subsequence of the weak-coupling expansion of the ground-state energy of the Yang-Gaudin model, that should read:

$$
\begin{aligned}
e_{F;\zeta(5)}(\gamma;\kappa=2) &= -\zeta(5)\sum_{n\geq 0}\frac{1}{2^{n+1}}\frac{n+2}{2n-1}\binom{2n}{n}\binom{\binom{n}{2}}{2}\frac{\gamma^{n+3}}{\pi^{2(n+2)}} \\
&= -\frac{15\zeta(5)}{4\pi^{10}}\gamma^6 - \frac{225\zeta(5)}{8\pi^{12}}\gamma^7 - \frac{2205\zeta(5)}{16\pi^{14}}\gamma^8 - \quad (23) \\
&\quad \frac{2205\zeta(5)}{4\pi^{16}}\gamma^9 - \frac{31185\zeta(5)}{16\pi^{18}}\gamma^{10} \dots
\end{aligned}
$$

This result is in full agreement with [51] to all published orders. As well as the former conjecture, this one has been confirmed up to order 50 by the authors of the aforementioned article [59]. The series converge provided that $\gamma \leq \pi^2/2$, and sums up to

$$
e_{F;\zeta(5)}(\gamma;\kappa=2) = -\frac{15}{4}\frac{\zeta(5)\gamma^6}{\pi^{10}}\frac{1-\frac{3}{2}\frac{\gamma}{\pi^2}+\frac{3}{4}\frac{\gamma^2}{\pi^4}}{\left[1-2\frac{\gamma}{\pi^2}\right]^{9/2}}. \quad (24)
$$

The limited radius of convergence of the series expansion in this conjecture and the previous one is due to the pole of the denominator in the functions involved in their resummation. At least one of the other subseries, most liktely the one that does not have any odd zeta prefactor, must have a radius of convergence equal to zero according to [51], where it was shown that the series on its whole has a null radius of convergence.

**Conjecture IV:** We noticed that the weak-coupling expansions of the ground-state energy in the case of the Yang-Gaudin and Lieb-Liniger model, along with the Yang-You theorem, suggest a general structure for the weak-coupling expansion of the ground-state energy of the $\kappa$-component model that reads:

$$
e_F(\gamma;\kappa) = \sum_{k\geq 0}\tilde{c}_k^F(\kappa)\pi^{2-k}\gamma^{k/2}, \quad (25)
$$

where $\tilde{c}_k^F$ is a linear, rational combination of 1, and products of zeta function evaluated at odd arguments, except for $\tilde{c}_2^F$ which also contains $\zeta(2)$. The fact that $\zeta(2)$ and more generally

$\zeta(2m)$ is specific to $\tilde{c}_2^F$ has been checked to low orders, and implemented by T. Reis and M. Marino in their calculations at very high orders to achieve workable computation times [59]. Equation (25) is a common generalization of Eqs. (19, 20).

In the balanced case, i.e. when the number of particles in each component is linked to the total number of particles by $N_i = N/\kappa$, it is known that [60]

$$e_F(\gamma;\kappa) = \frac{\pi^2}{3\kappa^2} + \frac{\kappa-1}{\kappa}\gamma + \dots \tag{26}$$

implying that $\tilde{c}_0^F(\kappa) = \frac{1}{3\kappa^2} \to_{\kappa \to +\infty} 0$, in agreement with the Lieb-Liniger result. It also means that for any spin, $\tilde{c}_1^F(\kappa) = 0$ and $\tilde{c}_2^F(\kappa) = \frac{\kappa-1}{\kappa}$. Furthermore, for odd orders, $\tilde{c}_{2j+1}^F(\kappa=2) = 0$.

# 4 Local correlation functions of the Lieb-Liniger Bose gas

In the Lieb-Liniger model, the M-body local correlation functions are usually defined as

$$g_M = \frac{\langle (\psi^\dagger(0))^M (\psi(0))^M \rangle}{n_0^M}, \tag{27}$$

where $\psi(x)$ is the annihilation operator of the bosons. These observables indicate the probability of observing $M$ particles at the same location at the same time. In a standard framework, their computation involves moments of the density of pseudo-momenta [36, 63],

$$e_{2k}(\gamma) = \frac{\int_{-1}^1 dy\, g(y, \alpha(\gamma)) y^{2k}}{[\int_{-1}^1 dy\, g(y, \alpha(\gamma))]^{2k+1}},$$

through the concept of connection, defined as a relationship between a local correlation function and coefficients of the short-distance expansion of the one-body correlation function [64]. This approach leads to rather complicated expressions for $g_M$ in terms of $e_{2k}$ and their derivatives, already at order $M = 3$ [65]. A few properies of the moments are given in appendix C. Based on strong-coupling expansions of $g_2$ and $g_3$ to order 20 in the inverse coupling constant, and inspired by the resummation of $e(\gamma)$ as in Eq. (13), we propose a conjecture about the general structure of the $M$-body correlation function.

**Conjecture V:** We guess that the structure of the strong-coupling expansion of $g_M(\gamma)$ takes the form:

$$g_M(\gamma) = \gamma^{M-1} \sum_{n=0}^{+\infty} \frac{\mathcal{G}_{n;M}(\gamma) \pi^{2n+M(M-1)}}{(2+\gamma)^{(M+1)(n+M-1)}},$$

where $\mathcal{G}_{n;M}$ is a polynomial of degree $(M-1)n$, and $G_{0;M} = \frac{1}{(2M-1)!!}\left(\frac{\prod_{j=1}^M j!}{\prod_{j=1}^{M-1}(2j-1)!!}\right)^2$. It is in agreement with our expansions of $g_2$ and $g_3$ to order 20, and with the most accurate explicit result to date [66] up to order $M^2 - M + 1$ in $1/\gamma$ for all values of $M$, in particular through the known expression of $G_{0,M}$. It also agrees with the exact result $g_1(\gamma) = 1$, on the condition that $\mathcal{G}_{n;0} = 0$ for $n \geq 1$.

Moreover, we have spotted out that $\mathcal{G}_{n;2} = \sum_{k=0}^n b_{k;n}\gamma^k$ with $b_{0;n} = 4a_{0;n}$ and $b_{n;n} = -(2n+1)a_{n-1;n}$ with the notations involved in Eq. (13) and below.

## 5  Pressure of the Lieb-Liniger Bose gas

Using an other formalism, the local correlation functions of the Lieb-Liniger model can be expressed in terms of solutions of other integral equations than the Lieb equation Eq. (3). For instance, the two-body local correlation function $g_2$ can be written as [67]

$$g_2(\gamma) = \frac{2}{\gamma}[e(\gamma) - p(\gamma)],$$

where by definition

$$p(\gamma) = \frac{1}{2\pi} \frac{\int_{-1}^{1} dz\, f(z; \alpha(\gamma))z}{[\int_{-1}^{1} dz\, g(z; \alpha(\gamma))]^3}.$$

The function $g(z; \alpha)$ has been defined above in Eq. (3), and $f(z; \alpha)$ is solution to the integral equation

$$f(z; \alpha) - \frac{1}{\pi} \int_{-1}^{1} dy \frac{\alpha}{\alpha^2 + (z - y)^2} f(y; \alpha) = z. \tag{28}$$

We combine several approaches to identify the physical meaning of $p(\gamma)$. It is a well-known fact that the Hellmann-Feynman theorem yields $e'(\gamma) = g_2$, where $'$ denotes the derivation with respect to $\gamma$ [36], thus

$$p = e - \frac{\gamma}{2} e'. \tag{29}$$

On the other hand, the pressure $P$ satisfies the thermodynamics relation [68]

$$P = \frac{2E_0}{L} - \frac{\hbar^2 n_0^3}{m} \frac{\gamma e'}{2}.$$

Therefore, we find out that

$$p = e - \frac{\gamma}{2} e' = \frac{P}{\hbar^2 n_0^2 / m}$$

is actually the dimensionless pressure of the Bose gas, an observable not quite studied so far in the case of the Lieb-Liniger model. More thermodynamic relations are derived in Appendix D. Our calculations of the strong-coupling expansion of the ground-state energy allowed us to obtain an expression for the pressure at the same order in $1/\gamma$, whose more general structure can be identified as in the case of the ground-state energy.

**Conjecture VI:**  Based on our strong-coupling expansion of $p(\gamma)$ at order 20 in the inverse coupling constant, we conjecture that it can be partially resummed as:

$$\frac{p(\gamma)}{\pi^2/3} = \gamma^3 \sum_{n=0}^{+\infty} \frac{\pi^{2n}}{(\gamma + 2)^{3n+3}} \mathcal{P}_n(\gamma), \tag{30}$$

where $\mathcal{P}_n$ are polynomials of degree $max(n-1,0)$, with rational coefficients of alternating signs. We have identified the first few polynomials as:

$$
\begin{aligned}
\mathcal{P}_0(\gamma) &= 1, \\
\mathcal{P}_1(\gamma) &= \frac{16}{3}, \\
\mathcal{P}_2(\gamma) &= -\frac{48}{5}\gamma + \frac{608}{45}, \\
\mathcal{P}_3(\gamma) &= \frac{768}{35}\gamma^2 - \frac{256}{5}\gamma + \frac{22336}{945}, \\
\mathcal{P}_4(\gamma) &= -\frac{512}{9}\gamma^3 + \frac{88064}{525}\gamma^2 - \frac{822272}{5775}\gamma + \frac{182784}{4725}, \\
\mathcal{P}_5(\gamma) &= \frac{12288}{77}\gamma^4 - \frac{27877376}{51975}\gamma^3 + \frac{429056}{693}\gamma^2 - \frac{1200128}{3465}\gamma + \frac{4698112}{93555}.
\end{aligned}
\tag{31}
$$

We found this structure using the property:

$$
\sum_{n=0}^{+\infty}\left(-\frac{2}{\gamma}\right)^n \binom{n+3k+2}{3k+2} = \frac{\gamma^{3(1-k)}}{(2+\gamma)^{3(k+1)}},
\tag{32}
$$

and in analogy with the structure of the ground-state energy, Eq. (15). Moreover, the highest-degree monomial of $\mathcal{P}_k$, where $k \geq 1$, seems to be affected of a coefficient

$$
p_{k;n} = 3 \times \frac{(-4)^{k+1}}{(2k+1)(k+2)}.
\tag{33}
$$

According to Eq. (29), the ground-state energy $e(\gamma)$ can be obtained from a differential equation knowing the pressure $p(\gamma)$, and the coefficients of their series expansion at weak and strong coupling are linked together. At weak coupling, the coefficient of $\gamma^2$ in the series expansion of the pressure is null, so the corresponding coefficient in the series expansion of $e_B(\gamma)$, that involves $\zeta(2)$, is actually an integration constant. This might explain why $\zeta(2)$ is specific to this order and does not appear at higher orders anymore. The differential equation also gives an explanation to the fact that products of zeta functions, such as $\zeta(3)^2$, appear in the expansion.

## 6   Conclusion and outlook

In conclusion, in this article we used the most accurate available data to guess patterns in the weak-coupling series expansion of the ground-state energy of the Yang-Gaudin model. We proposed two conjectures about the mathematical structure of subseries whose terms share a common prefactor that involves $\zeta(3)$ and $\zeta(5)$ respectively. Although they encapsulate an infinite number of terms, they do not encompass multiples of zeta functions evaluated at odd arguments, that appear at higher orders. Lack of data did not allow us to generalize our conjectures to all odd arguments of the zeta function, but the expressions already obtained may serve as a guideline to identify more coefficients in the future. The subseries we have resummed have a finite radius of convergence, and are not responsible for the null radius of convergence of the full series, which should be understood from the subseries devoid of odd zeta terms.

All the results obtained for the ground-state energy of the Yang-Gaudin model, $e_F(\gamma;\kappa=2)$, readily yield their equivalent in the super-Tonks-Girardeau regime of the Lieb-Liniger model,

through the relation $e_B^{STG}(\gamma) = 16 e_F\left(\frac{|\gamma|}{4}\right)$. Surprisingly, the coefficients of the standard Lieb-Liniger model with repulsive interactions seem more difficult to guess than those of the Yang-Gaudin model.

We also made conjectures about the structures of the weak-coupling and strong-coupling series expansion of the ground-state energy of $\kappa$-component fermions. The fact that almost all weak-coupling coefficients are rational linear combinations of 1 and zeta function evaluated at odd arguments should be related to the model's integrability [59]. The only exception here concerns the coefficient of $\gamma^2$, which might be a special feature of the $\kappa$-component model in general. Its role of an integration constant may be a clue to this special behavior. We note that a similar structure has been observed in the emptiness formation probability of the XXX spin chain [69], where the link between the appearance of the zeta function at odd arguments and integrability has been investigated and a similar conjecture has been stated. In this case, however, factors of ln(2) also appear, that are absent in the weak-coupling expansions of the ground-state energy of the Lieb-Liniger and Yang-Gaudin models. However, both factors ln(2) and $\zeta(2j + 1)$ can be encompassed in the more general framework of polylogarithms. Understanding at a deeper level the link between number theory and integrability is a thrilling problem in mathematical physics.

Then, we devoted our attention to the local correlation functions of the Lieb-Liniger model. Inspired by a conjecture about the strong-coupling expansion of the ground-state energy of the Lieb-Liniger model, and using series expansions of the two- and three-body correlation functions, we proposed yet another conjecture about the structure of general local correlation functions, that encompasses the most accurate analytical results available in the literature as special cases.

Finally, we combined two approaches to evaluate the two-body correlation function, to identify another ground-state observable as the pressure of the Lieb-Liniger Bose gas. We proposed a conjecture about the structure of its strong-coupling expansion.

As an outlook, it would be interesting to systematically compare several approaches to the local correlation functions. In [64], we pointed out the existence of a general framework through the notion of connection, that would deserve more investigations. In parallel, an other very interesting formalism exists, whose solution requires to solve several integral equations. It is thus more demanding in terms of computations, but better adapted to numerical studies as it does not involve derivatives. Its structure is simpler, and explicitly known at all orders $M$ [70], it is valid at finite temperature and can be adapted to out-of-equilibrium situations, yet actual calculations of series expansions in terms of the coupling constant are still lacking.

As far as non-local correlation functions of the Lieb-Liniger model are concerned, they have been studied extensively in the classical textbook [71]. Working out explicitly their dependence on the coupling constant and temperature is still a partially-open problem.

## Acknowledgments

We thank Jean Decamp for driving our attention on ref. [50], Anna Minguzzi for welcoming us at LPMMC in the final stage of writing, Maxim Olshanii for his interest in conjecture V, and Tomás Reis for useful comments and insights on conjectures II, III and IV.

## A Strong-coupling expansion of the ground-state energy of the Lieb-Liniger model

In this Appendix, we give the explicit expansion of the dimensionless ground-state energy in the inverse coupling constant in the vicinity of the Tonks-Girardeau regime $\gamma \to +\infty$:

$$
\begin{aligned}
\frac{e(\gamma)}{\pi^2/3} = {}& 1 - \frac{4}{\gamma} + \frac{12}{\gamma^2} - \frac{32}{\gamma^3} + \frac{80}{\gamma^4} - \frac{192}{\gamma^5} + \frac{448}{\gamma^6} - \frac{1024}{\gamma^7} + \frac{2304}{\gamma^8} - \frac{5120}{\gamma^9} \\
& + \frac{11264}{\gamma^{10}} - \frac{24576}{\gamma^{11}} + \frac{53248}{\gamma^{12}} - \frac{114688}{\gamma^{13}} + \frac{245760}{\gamma^{14}} - \frac{524288}{\gamma^{15}} + \frac{1114112}{\gamma^{16}} - \frac{2359296}{\gamma^{17}} \\
& + \frac{4980736}{\gamma^{18}} - \frac{10485760}{\gamma^{19}} + \frac{22020096}{\gamma^{20}} \\
& + \frac{\pi^2}{\gamma^3}\left(\frac{32}{15} - \frac{64}{3\gamma} + \frac{128}{\gamma^2} - \frac{1792}{3\gamma^3} + \frac{7168}{3\gamma^4} - \frac{43008}{5\gamma^5} + \frac{28672}{\gamma^6} - \frac{90112}{\gamma^7} + \frac{270336}{\gamma^8}\right) \\
& + \frac{\pi^2}{\gamma^3}\left(-\frac{2342912}{3\gamma^9} + \frac{32800768}{15\gamma^{10}} - \frac{5963776}{\gamma^{11}} + \frac{47710208}{3\gamma^{12}} - \frac{124780544}{3\gamma^{13}}\right) \\
& + \frac{\pi^2}{\gamma^3}\left(\frac{106954752}{\gamma^{14}} - \frac{1354760192}{5\gamma^{15}} + \frac{677380096}{\gamma^{16}} - \frac{1673527296}{\gamma^{17}}\right) \\
& + \frac{\pi^4}{\gamma^5}\left(-\frac{96}{35} + \frac{2096}{45\gamma} - \frac{19712}{45\gamma^2} + \frac{15104}{5\gamma^3} - \frac{51200}{3\gamma^4} + \frac{1255936}{15\gamma^5} - \frac{1847296}{5\gamma^6}\right) \\
& + \frac{\pi^4}{\gamma^5}\left(\frac{157560832}{105\gamma^7} - \frac{85516288}{15\gamma^8} + \frac{20500480}{\gamma^9} - \frac{3167617024}{45\gamma^{10}} + \frac{10457055232}{45\gamma^{11}}\right) \\
& + \frac{\pi^4}{\gamma^5}\left(-\frac{3707764736}{5\gamma^{12}} + \frac{34461712384}{15\gamma^{13}} - \frac{145636720640}{21\gamma^{14}} + \frac{102284394496}{5\gamma^{15}}\right) \\
& + \frac{\pi^6}{\gamma^7}\left(\frac{512}{105} - \frac{20224}{175\gamma} + \frac{198784}{135\gamma^2} - \frac{12669184}{945\gamma^3} + \frac{6161408}{63\gamma^4} - \frac{574306304}{945\gamma^5}\right) \\
& + \frac{\pi^6}{\gamma^7}\left(\frac{2254759936}{675\gamma^6} - \frac{5246365696}{315\gamma^7} + \frac{72470953984}{945\gamma^8} - \frac{8921808896}{27\gamma^9}\right) \\
& + \frac{\pi^6}{\gamma^7}\left(\frac{141315080192}{105\gamma^{10}} - \frac{24676999233536}{4725\gamma^{11}} + \frac{18362588987392}{945\gamma^{12}} - \frac{3135253774336}{45\gamma^{13}}\right) \\
& + \frac{\pi^8}{\gamma^9}\left(-\frac{1024}{99} + \frac{492032}{1575\gamma} - \frac{124928}{25\gamma^2} + \frac{9857536}{175\gamma^3} - \frac{7534592}{15\gamma^4} + \frac{11315200}{3\gamma^5}\right) \\
& + \frac{\pi^8}{\gamma^9}\left(-\frac{5577834496}{225\gamma^6} + \frac{32942620672}{225\gamma^7} - \frac{138548740096}{175\gamma^8} + \frac{35774136320}{9\gamma^{10}}\right) \\
& + \frac{\pi^8}{\gamma^9}\left(-\frac{1179996127232}{63\gamma^{10}} + \frac{22973599973376}{275\gamma^{11}}\right) \\
& + \frac{\pi^{10}}{\gamma^{11}}\left(\frac{24576}{1001} - \frac{29679616}{33075\gamma} + \frac{4472756224}{259875\gamma^2} - \frac{3995396096}{17325\gamma^3} + \frac{226836987904}{93555\gamma^4}\right) \\
& + \frac{\pi^{10}}{\gamma^{11}}\left(\frac{226836987904}{93555\gamma^5} - \frac{1993002090496}{93555\gamma^6} + \frac{8455393705984}{51975\gamma^7}\right) \\
& + \frac{\pi^{10}}{\gamma^{11}}\left(\frac{7509802968940544}{1091475\gamma^8} - \frac{409672538914816}{10395\gamma^9}\right)
\end{aligned}
$$

$$
\begin{aligned}
&+\frac{\pi^{12}}{\gamma^{13}}\left(-\frac{4096}{65}+\frac{94450688}{35035\gamma}-\frac{1422291795968}{23648625\gamma^2}+\frac{22033051721728}{23648625\gamma^3}\right) \\
&+\frac{\pi^{12}}{\gamma^{13}}\left(-\frac{17753139208192}{1576575\gamma^4}+\frac{4815327978459136}{42567525\gamma^5}\right. \\
&\qquad\qquad\left.-\frac{643580495888384}{654885\gamma^6}+\frac{76876913487282176}{10135125\gamma^7}\right) \\
&+\frac{\pi^{14}}{\gamma^{15}}\left(\frac{131072}{765}-\frac{1132822528}{135135\gamma}+\frac{18440364032}{86625\gamma^2}-\frac{3976857599787008}{1064188125\gamma^3}\right) \\
&+\frac{\pi^{14}}{\gamma^{15}}\left(\frac{433893767446528}{8513505\gamma^4}-\frac{1166349005750272}{2027025\gamma^5}\right) \\
&+\frac{\pi^{16}}{\gamma^{17}}\left(-\frac{786432}{1615}+\frac{717866991616}{26801775\gamma}-\frac{32984804753408}{43295175\gamma^2}+\frac{1765201175773184}{118243125\gamma^3}\right) \\
&+\frac{\pi^{18}}{\gamma^{19}}\left(\frac{2097152}{1463}-\frac{23334092275712}{266741475\gamma}\right) \\
&+O\left(\frac{1}{\gamma^{21}}\right).
\end{aligned}
\tag{34}
$$

## B  The even-odd trick

To seek the structure of the ground-state energy, in this appendix we rely on the fact that any function can be decomposed in a unique way as a sum of an even and an odd function. The dimensionless ground-state energy is thus split into

$$
e_B(\gamma) = e_{even}(\gamma) + e_{odd}(\gamma). \tag{35}
$$

We try and identify patterns of increasing complexity in $e_{even}$ and $e_{odd}$, writing

$$
e_{even/odd}(\gamma) = \sum_{n=0}^{+\infty} e_{even/odd,n}(\gamma). \tag{36}
$$

In doing so, we found

$$
e_{even,0}(\gamma) = \frac{\pi^2}{3}\sum_{n=0}^{+\infty}\binom{2n+1}{1}\left(\frac{2}{\gamma}\right)^{2n}, \tag{37}
$$

$$
e_{even,1}(\gamma) = -\frac{4}{45}\pi^4\sum_{n=0}^{+\infty}\binom{2n+5}{4}\left(\frac{2}{\gamma}\right)^{2n+4}, \tag{38}
$$

and for the odd function:

$$
e_{odd,0}(\gamma) = -\frac{\pi^2}{3}\sum_{n=0}^{+\infty}\binom{2n+2}{1}\left(\frac{2}{\gamma}\right)^{2n+1}, \tag{39}
$$

and

$$
e_{odd,1}(\gamma) = \frac{4}{45}\pi^4\sum_{n=0}^{+\infty}\binom{2n+4}{4}\left(\frac{2}{\gamma}\right)^{2n+3}, \tag{40}
$$

yet at higher orders the structures suddenly complexify.

## C  Moments of the density of pseudo-momenta

This appendix is devoted to the asymptotic properties of the integral

$$I_{2k}(\gamma) = \int_{-1}^{1} dy\, g(y; \alpha(\gamma)) y^{2k}, \tag{41}$$

involved in the calculation of the moments of the density of pseudo-momenta, Eq. (28).

Using a strong-coupling expansion, we found out that for $k = 0$, $I_0$ involves the series expansion of

$$\frac{1}{\pi}\left(\frac{\gamma}{2+\gamma}\right)^{-1} \tag{42}$$

and

$$-\frac{4\pi}{3\gamma^3}\sum_{n=0}^{+\infty}(n+1)\left(-\frac{2}{\gamma}\right)^n = -\frac{4\pi}{3\gamma^3}\left(\frac{\gamma}{2+\gamma}\right)^2. \tag{43}$$

For $k = 1$, $I_2$ involves

$$\frac{1}{3\pi}\left(\frac{\gamma}{2+\gamma}\right)^{-1} \tag{44}$$

and

$$-\frac{28\pi}{45\gamma^3}\sum_{n=0}^{+\infty}(n+1)\left(-\frac{2}{\gamma}\right)^n = -\frac{28\pi}{45\gamma^3}\left(\frac{\gamma}{2+\gamma}\right)^2. \tag{45}$$

We thus guessed that $I_{2k}$ contains the following structures:

$$\frac{1}{\pi}\frac{1}{(2k+1)}\left(\frac{\gamma}{2+\gamma}\right)^{-1} \tag{46}$$

and

$$-\frac{4\pi}{3\gamma^3}\left(\frac{4(k+1)-1}{4(k+1)^2-1}\right)\left(\frac{\gamma}{2+\gamma}\right)^2. \tag{47}$$

At higher orders the structures suddenly become more complex, for instance we identified

$$\frac{16\pi^3}{5\gamma^6}\frac{4k^2+11k+5}{8k^3+36k^2+46k+15}\left(\frac{\gamma}{2+\gamma}\right)^5(2\gamma + a(k)), \tag{48}$$

where $a(k)$ is a polynomial function of the variable $k$.

We also noticed that

$$\frac{1}{2k+1} = \frac{1}{1(k+1)+k} \tag{49}$$

and

$$\frac{4(k+1)-1}{4(k+1)^2-1} = \frac{4k+3}{(4k+3)(k+1)+k}, \tag{50}$$

which might be the correct way to write prefactors to identify a general structure.

# D More relationships for the pressure

The dimensionless pressure $p$ satisfies

$$p = e - \frac{\gamma}{2} e'. \tag{51}$$

It is known that the dimensionless chemical potential satisfies [1]

$$\mu = 3e - \gamma e'. \tag{52}$$

Combining these expressions yields, in real units,

$$p = \frac{1}{2}(\mu - e) = \frac{1}{2n_0^2}\left[\frac{\partial E_0}{\partial N} - \frac{E_0}{N}\right] = \frac{N}{2n_0^2}\frac{\partial}{\partial N}\left(\frac{E_0}{N}\right). \tag{53}$$

Also, introducing the operators

$$F : f \rightarrow 3f - \gamma f' \tag{54}$$

and

$$G : g \rightarrow g - \frac{\gamma}{2} g', \tag{55}$$

we notice that

$$p = G(e) \tag{56}$$

and

$$\mu = F(e). \tag{57}$$

Since $F$ and $G$ commute,

$$GoF(e) = FoG(e), \qquad F(p) = G(\mu),$$

and thus

$$3p - \gamma p' = \mu - \frac{\gamma}{2}\mu'. \tag{58}$$

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
