# Peer review of "Conjectures about the structure of strong- and weak-coupling expansions of a few ground-state observables in the Lieb-Liniger and Yang-Gaudin models"

_SciPost Physics, doi:SciPost Phys. 7, 055 (2019)_

## Round 1 · Referee Report · Anonymous (Referee 1) · 2019-8-13

Strengths

Important models. Attempt to relate integrability to number theory.

Weaknesses

The author do not know that the most important in correlation functions is space
and time dependence.
The author do not know the literature on the subject.

Report

1) The relation of integrability and number theory was explained in the paper Quantum Correlations and Number Theory by H. E. Boos, V. E. Korepin, Y. Nishiyama, M. Shiroishi, Journal of Physics A Math. and General, vol 35, pages 4443-4452, 2002

2) Correlation functions were calculated in Lieb-Liniger model in the tex book Quantum Inverse Scattering Method and Correlation Functions by V.E. Korepin, N.M. Bogoliubov and A.G. Izergin, Cambridge University Press , 1993.

Requested changes

The author has to compare his results to the literature [see report] and resubmit the paper.

  • validity: good
  • significance: good
  • originality: ok
  • clarity: good
  • formatting: good
  • grammar: good

Author:  Guillaume Lang  on 2019-08-19  [id 580]

(in reply to Report 1 on 2019-08-13)

We thank the referee for pointing out these two relevant references, and apologize for omitting them. They have been added to the bibliography in next version of the article.

We are fully aware that 'the most important in correlation functions is space
and time dependence', and have already published several papers on this topic. However, they are beyond the scope of this very article, and we do not wish to address them here. A few comments on their importance have been added, though, and they are now mentioned in outlook.

The conjecture in reference 1) suggested by the reviewer is indeed very similar to ours, though both the model and observable differ. The appearance of ln(2) in one case and not the other is an important fact, that we had not envisaged at first. Comments have been added to the conjecture in accordance.

The results in reference 2) are exact and very important for mathematical physics, but not explicit beyond first order. It is thus more relevant to compare ours to the most accurate expansions available to date: the exact result for g1, order 20 in 1/gamma for g2 and g3 and first non-zero order for gM.

Resubmission will follow soon.

---

## Round 1 · Referee Report · Anonymous (Referee 2) · 2019-8-26

Strengths

Timely paper supplementing nicely very recent results on the subject.

Weaknesses

The conjectures given in the paper only address very specific contributions to the groundstate energy without any particular physical significance.

Report

The author studies the groundstate energy for the celebrated Lieb-Liniger and Yang-Gaudin models, which are integrable systems of Bosons and Fermions in one dimension with $\delta$ interaction of strength $\gamma$. Known exact results involving integral equations have been derived more than 50 years ago. While the strong coupling expansion $\gamma\to\infty$ can be easily extracted in a systematic way from the integral equations, the weak coupling expansions $\gamma\to0$ are significantly harder, with only the very first few terms known before a systematic approach was devised [50,51] in 2019.

Although systematic algorithms now exist in some cases for both strong and weak coupling expansions, there is no closed formula for either expansion up to arbitrary order. The author was able to formulate plausible conjectures about the structure of these expansions, and even perform a complete resummation of some portions of the expressions for the groundstate energy, paving the way to a possible combinatorial solution for the full expansions.

Requested changes

1) It would be useful to add a discussion of the radius of convergence of the strong and weak coupling expansions. Can they be explained by the location of poles in your various resummations ? This would give a clearer motivation for the paper.

2) At several places, e.g. in conjecture V when comparing to [66], it is not clear up to which order the conjectures have been checked. This must be said more explicitly.

3) Maybe you could comment on the advantages of the algorithm from [40] for the strong coupling expansion, if any, compared with the more standard inversion of the integral operator as explained in e.g. [46].

4) Equation (19): it should be $O(\gamma^{(n+1)/2})$ and not $O(\gamma^{(k+1)/2})$.

5) Below (12), the standard notation for the floor function is $\lfloor ... \rfloor$.

  • validity: good
  • significance: ok
  • originality: good
  • clarity: high
  • formatting: good
  • grammar: good

Author:  Guillaume Lang  on 2019-08-28  [id 586]

(in reply to Report 2 on 2019-08-26)

We answer the referee's queries point by point:

1) It would be useful to add a discussion of the radius of convergence of the strong and weak coupling expansions. Can they be explained by the location of poles in your various resummations ? This would give a clearer motivation for the paper. -> The strong-coupling expansion in its standard form is valid for \alpha>2, i.e. for \gamma>4.53. Once partially resummed, the radius of convergence becomes infinite. -> The radius of convergence of the weak-coupling expansion is readily understood from the poles of the partial resummations, as pointed out by the referee. -> Comments on these points have been added.

2) At several places, e.g. in conjecture V when comparing to [66], it is not clear up to which order the conjectures have been checked. This must be said more explicitly. -> Details have been provided more explicitly everywhere in the new version.

3) Maybe you could comment on the advantages of the algorithm from [40] for the strong coupling expansion, if any, compared with the more standard inversion of the integral operator as explained in e.g. [46]. -> Already at third order, the terms of the Neumann series are too complicated, thus they have never been computed analytically. The method of ref.[40] yields exact coefficients at very high orders in comparison.

4) Equation (19): it should be O(γ(n+1)/2) and not O(γ(k+1)/2). -> Modified as requested.

5) Below (12), the standard notation for the floor function is ⌊...⌋ -> Modified as requested.

---

## Round 1 · Referee Report · Anonymous (Referee 3) · 2019-9-11

Strengths

1) Provides an interesting set of conjectures on the weak and strong coupling expansions of two important integrable systems in many-body theory.

2) These conjectures might stimulate further research and improve the analytic study of the model.

Weaknesses

The results of this paper do not seem to have a strong bearing on the physics of these models.

Report

In this paper, a series of conjectures are made on the structure of the weak and strong coupling expansions of the ground state energy in the Lieb-Liniger and the Gaudin-Yang models, as well as on other observables, based on the most up to date analytic results for these expansions. The author had already made an interesting contribution to the subject in his Ph.D. by identifying some numerical results in terms of analytic expressions featuring zeta functions, and now, based on these recent results, he presents some guesses about partial structures in these expansions which have been validated to very high order. Although these conjectures are somewhat "numerological", and they do not lead to strong intuitions on the underlying physics, they might be useful in future studies of these models.

Requested changes

In his reply to another referee report, the author says that "The radius of convergence of the weak-coupling expansion is readily understood from the poles of the partial resummations, as pointed out by the referee". This seems to imply that the radius of convergence is finite. However, this contradicts one of the main findings of refs. [50,51], which have shown numerically that the weak-coupling expansion diverges factorially (as in many other quantum-mechanical models) and therefore has zero radius of convergence. In view of this, the author should address this issue more carefully.

---

## Round 4 · Referee Report · Anonymous (Referee 1) · 2019-10-10

Strengths

The author try to relate number theory with dynamical systems

Weaknesses

Conjectures are not proven. Only two dynamical systems aRE CONSIDERED

Report

Afte resubmission the paper became a little better.

---

## Round 4 · Referee Report · Anonymous (Referee 2) · 2019-10-10

Report

The reply by the author and the changes made to the paper address satisfactorily the points raised in my previous report. I think the paper is now suitable for publication in scipost.

---

## Round 4 · Referee Report · Anonymous (Referee 3) · 2019-10-12

Report

The author has clarified various points. I think the article can be recommended for publication

---

## Editorial Decision

published